# Do Stochastic Parrots have Feelings Too?
## Improving Neural Detection of Synthetic Text via Emotion Recognition

**Alan Cowap**[1] and **Yvette Graham**[2] and **Jennifer Foster**[3]

[1,3]School of Computing, Dublin City University
[1]SFI Centre for Research Training in Machine Learning at Dublin City University
`alan.cowap2@mail.dcu.ie, jennifer.foster@dcu.ie`
[2]School of Computer Science and Statistics, Trinity College Dublin
`ygraham@tcd.ie`

## Abstract

Recent developments in generative AI have shone a spotlight on high-performance synthetic text generation technologies. The now wide availability and ease of use of such models highlights the urgent need to provide equally powerful technologies capable of identifying synthetic text. With this in mind, we draw inspiration from psychological studies which suggest that people can be driven by emotion and encode emotion in the text they compose. We hypothesize that pretrained language models (PLMs) have an *affective deficit* because they lack such an emotional driver when generating text and consequently may generate synthetic text which has *affective incoherence* i.e. lacking the kind of emotional coherence present in human-authored text. We subsequently develop an emotionally aware detector by fine-tuning a PLM on emotion. Experiment results indicate that our emotionally-aware detector achieves improvements across a range of synthetic text generators, various sized models, datasets, and domains. Finally, we compare our emotionally-aware synthetic text detector to ChatGPT in the task of identification of its own output and show substantial gains, reinforcing the potential of emotion as a signal to identify synthetic text. Code, models, and datasets are available at `https://github.com/alanagiasi/emoPLMsynth`

## 1 Introduction

Modern PLMs can surpass human-level baselines across several tasks in general language understanding (Wang et al., 2018, 2019) and can produce synthetic text that can exceed human level quality, such as synthetic propaganda thought to be more plausible than human written propaganda (Zellers et al., 2019). PLMs have been used to generate disinformation (Zellers et al., 2019; Brown et al., 2020), left- or right-biased news (Gupta et al., 2020), fake comments (Weiss, 2019), fake reviews (Adelani et al., 2019), and plagiarism (Gao et al., 2022) and

can generate synthetic text at scale, across domains, and across languages.

The increasing high quality of synthetic text from larger and larger PLMs brings with it an increasing risk of negative impact due to potential misuses. In this work, we focus on the task of synthetic text detection. Due to the potentially profound consequences of global synthetic disinformation we focus mainly, but not exclusively, on the detection of synthetic text in the news domain.[1] Synthetic news has already been published on one highly reputable media website, only later to be withdrawn and apologies issued for the "breach of trust" (Crowley, 2023a,b).

Current approaches to synthetic text detection tend to focus on learning artefacts from the output distribution of PLMs (Gehrmann et al., 2019; Pillutla et al., 2021; Mitchell et al., 2023), e.g. increased perplexity caused by nucleus sampling (Zellers et al., 2019). However, PLM distributions are dependent on training data and numerous hyperparameter choices including model architecture and sampling strategy. This gives rise to a combinatorial explosion of possible distributions and makes the task of synthetic text detection very difficult. Furthermore, it is not unexpected that performance decreases when classifying out-of-distribution instances, and there is a growing field of work investigating this shortcoming (Yang et al., 2023).

In this work, we consider not only the PLM output distribution, but also the other side of the synthetic text detection coin – human factors. We present a novel approach to the task of synthetic text detection which aims to exploit any difference between expression of emotion in human-authored and synthetic text. Neural word representations can have difficulty with emotion words, and PLM sampling strategies are stochastic rather than driven by emotion – we use the term *affective deficit* to refer

---

[1]The news domain is recognised as having high emotional content (Strapparava and Mihalcea, 2007; Bostan et al., 2020).

to these shortcomings. Thus, the resulting synthetic text can express emotion in an incoherent way, and we introduce the term *affective incoherence* to refer to this type of limitation. To be clear, we do not contend that synthetic text is devoid of emotion, rather that the emotional content of synthetic text may be affectively incoherent, and that this affective incoherence stems from the underlying affective deficit of the PLM.

For the purpose of demonstration of the affective deficit that we believe to be characteristic of text produced by PLMs, we provide the following simple example of human- versus machine-authored text with positive emotion words highlighted in orange and negative emotion words in pink. One shows coherent emotion expected of human-authored text, while the other demonstrates affective incoherence (see footnote[2] to reveal which was synthetic/human-authored text).

1. *Roberts chuckled when asked if he was happy to be on the other team now when Puig's name comes up. "Yeah, I am happy," he said, smiling.*
2. *I'm really happy for him. Over the course of those three seasons, the 25-year-old has gone from rolling to poor to worse and old.*

In this simple example, we have demonstrated one kind of affective incoherence present in synthetic text but we suspect that fine-tuning an emotionally-aware PLM could detect additional and more complex emotional patterns that might go undetected by humans. We hypothesise that the *affective deficit* of PLMs could result in synthetic text which is *affectively incoherent*, which could be useful in distinguishing it from human text.

We use a transfer learning (Pan and Yang, 2010) method to train an "emotionally-aware" detector model. By fine-tuning a PLM first on emotion classification and then on our target task of synthetic text detection, we demonstrate improvements across a range of synthetic text generators, various sized models, datasets and domains. Furthermore, our emotionally-aware detector proves to be more accurate at distinguishing between human and ChatGPT text than (zero-shot) ChatGPT itself.

Finally, we create two new datasets: *NEWSsynth*, a dataset of 20k human and synthetic news articles, and *ChatGPT100*, a testset of 100 human and ChatGPT texts on a range of topics. We make all code,

models and datasets publicly available to aid future research.[3]

## 2  Related Work

People are relatively poor at detecting synthetic text, and have been shown to score just above random chance (Gehrmann et al., 2019; Uchendu et al., 2021). Hybrid systems, such as GLTR (Gehrmann et al., 2019) for example, use automation to provide information to aid human classification, highlighting a text sequence using colours to represent likeness to the PLM output distribution such as GPT-2 (Radford et al., 2019). Gehrmann et al. (2019) reported an increase in detection accuracy of approximately 18% (from 54% to 72%) using GLTR, while Uchendu et al. (2021) report an F1 score of 46% using GLTR with a heuristic based on an analysis of human text.

Both human and hybrid approaches involve human decisions, which can be slow, expensive, susceptible to bias, and inconsistent. Automatic detection produces the best results for synthetic text detection. This usually involves training PLMs to detect other PLMs, but zero-shot detection methods also exist, e.g. DetectGPT (Mitchell et al., 2023). Potentially the best supervised detector, BERT, can detect synthetic text from 19 different generators with a mean F1 of 87.99%, compared to 56.81% for hybrid, and worst of all humans at 53.58% (Uchendu et al., 2021).

Performance of SOTA detectors can however be inconsistent and unpredictable due to several factors specific to both the detector and generator, including: model size and architecture, training data and domain thereof, sampling strategy, hyperparameter selection, and sentence length. As mentioned above, Uchendu et al. (2021) showed the best of these models (BERT) achieves a mean F1 of 87.99% on 19 different synthetic text generators. However, the mean score hides the wide range (≈53%) of F1's, ranging from as low as 47.01% to 99.97%, for distinct synthetic text generators. This volatility may be due in part to the detector simply learning artefacts of the generator distribution. Consequently, the task of synthetic text detection is somewhat of an arms race with detectors playing catch-up, forced to learn ever-changing distributions due to the numerous factors that can potentially change those distributions.

Existing approaches to synthetic text detection

---

[2](1) is human-authored while (2) is synthetic text. Both are from the *NEWSsynth* dataset (see §4.2).

[3]https://github.com/alanagiasi/emoPLMsynth

exploit properties of synthetic text. Synthetic text can be incoherent and degrade as the length of generated text increases (Holtzman et al., 2020), perplexity increases with increasing length unlike human text (Zellers et al., 2019), and PLMs are susceptible to sampling bias, induction bias, and exposure bias (Ranzato et al., 2016). For example, exposure bias can contribute to brittle text which is repetitive, incoherent, even containing hallucinations (Arora et al., 2022). Synthetic text can have an inconsistent factual structure, such as mentioning irrelevant entities (Zhong et al., 2020). Perhaps unsurprisingly, synthetic text detection is less difficult with longer excerpts of generated text, for both humans and machines (Ippolito et al., 2020).

One aspect of writing that has not, up to now, been a focus of synthetic text detection efforts is the expression of emotion. The problem of encoding emotion was first identified in neural NLP with static embeddings such as word2vec (Mikolov et al., 2013; Wang et al., 2020a). Static word embeddings have difficulty distinguishing antonymns from synonyms (Santus et al., 2014). This deficit is present in embeddings for words which represent opposing emotions (e.g. joy-sadness) (Seyeditabari and Zadrozny, 2017). Furthermore, words representing opposing emotions can have closer embeddings relative to words representing similar emotions (Agrawal et al., 2018). There have been various approaches to address this affective deficit in embeddings, such as transfer learning from sentiment analysis (Kratzwald et al., 2018), an additional training phase using an emotional lexicon and psychological model of emotions (Seyeditabari et al., 2019), and combining separately-learned semantic and sentiment embedding spaces (Wang et al., 2020a).

Addressing potential affective deficits of PLMs is also the goal of work aiming to make dialogue systems more empathetic. For example Huang et al. (2018) force dialogue generation to express emotion based on the emotion detected in an utterance, while Rashkin et al. (2019) follow a similar approach with a transformer architecture to make the system more empathetic. In contrast, Wang et al. (2020b) report that human text can display consistency in emotional content whereby similar emotions tend to occur adjacent to each other while dissimilar emotions seldom do.[4]

---

[4] For a comprehensive survey of sentiment control in synthetic text see (Lorandi and Belz, 2023) and for studies of emotion in human writing, see (Brand, 1985, 1987, 1991;

Past work in synthetic text detection has focused on the properties of synthetic text generators and is yet to take advantage of the factors that potentially influence human-authored text, such as the emotions humans express in the text they write. Our work exploits this PLM affective deficit to improve synthetic text detection.

## 3 Equipping PLMs with Emotional Intelligence

Our method is illustrated in Figure 1. The process works as follows:

1. PLMSYNTH: In the leftmost column of Figure 1, human articles and synthetic articles are used to fine-tune a PLM to discriminate between the two kinds of text. This is indicated by the blue nodes in the PLM illustration.
2. EMOPLM: In the middle column of Figure 1, a second dataset annotated for emotions with Ekman's 6 emotions (Ekman, 1992, 1999, 2016) is used to fine-tune a PLM on the task of emotion classification. This makes our model emotionally-aware, as indicated by the red nodes in the PLM illustration.
3. EMOPLMSYNTH: The multi-class (6 head) classification layer from emoPLM is removed and replaced with a binary classification layer. The emotionally-aware PLM is then fine-tuned on the task of discriminating between human and synthetic articles. The PLM is still emotionally-aware while also being able to detect synthetic text - as indicated by the red and blue nodes respectively in the PLM.

We conduct experiments using various PLM sizes, architectures, datasets, and domains for synthetic text generation and detection.

## 4 News Domain Experiments

### 4.1 Generator and Detector Models

To generate synthetic text, we use the Grover causal PLM (GPT-2 architecture) pretrained on 32M news articles from the RealNews dataset (Zellers et al., 2019). We choose BERT (Devlin et al., 2019) as our main detector model since it is freely available and performs well in several tasks including sequence classification. A baseline BERT model (we call this BERTsynth) is fine-tuned on the task of

---

Bohn-Gettler and Rapp, 2014; Knaller, 2017).

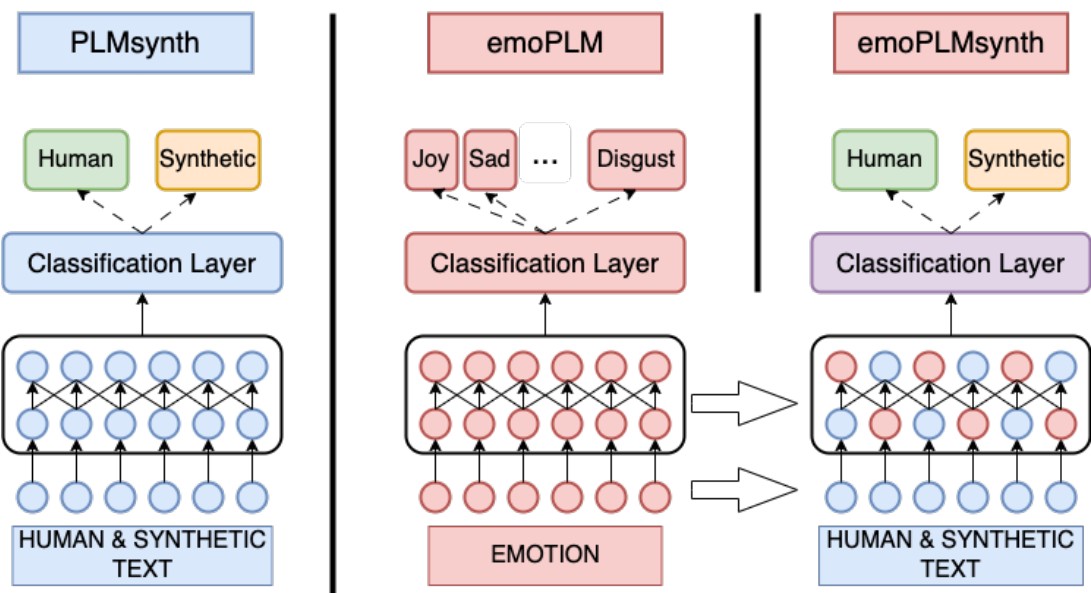

Figure 1: The emotionally-aware PLM (emoPLMsynth) takes advantage of its prior fine-tuning on emotion to improve performance on the task of synthetic text detection. In contrast, the standard PLM fine-tuned only on synthetic text detection (PLMsynth) has no training on emotion. Our experiments show the emotionally-aware PLM (emoPLMLsynth) outperforms the standard PLM (PLMsynth) in multiple scenarios.

synthetic text detection, while our proposed model is the same BERT model, firstly fine-tuned on emotion classification (we call this intermediate model emoBERT) before further fine-tuning for synthetic text detection. This final proposed model is referred to as emoBERTsynth.

## 4.2 Datasets

We create and release *NEWSsynth*, a dataset containing 10k human and 10k synthetic news articles. 10k human-authored news articles were taken from the RealNews-Test dataset (Zellers et al., 2019) and used as a prompt to Grover_base to generate a corresponding 10k synthetic articles. The prompt includes the news article, headline, date, author, web domain etc. as described by Zellers et al. (2019). The dataset was split 10k-2k-8k for train, validation, and test respectively, the same ratio used by Zellers et al. (2019) with 50:50 human:synthetic text in each split, see Appendix B.3 for details. An investigation of length of human vs synthetic text is provided in Appendix E.

In a second experiment, we also use the full RealNews-Test dataset itself, which comprises the same 10k human news articles used in *NEWSsynth* and 10k synthetic articles generated by Grover_mega. The use of synthetic text generated by Grover_mega instead of Grover_base allows comparison of BERTsynth and emoBERTsynth on text

generated by a larger generator model, and against results reported for other models on this dataset.

We use the GoodNewsEveryone dataset (Bostan et al., 2020) to train emoBERT. This dataset contains 5k news headlines, and was chosen since it is within the target domain (news) and language (English) and is annotated with categorical emotions. The 15 emotion labels from GoodNewsEveryone were reduced to 11 emotions using the mapping schema of (Bostan and Klinger, 2018), and further reduced to 6 emotions based on the Plutchik Wheel of Emotion (Plutchik, 1980, 2001) – see Table 1 and Figure 3 in Appendix A – resulting in 5k news headlines labelled with Ekman's 6 basic emotions, the most frequently used categorical emotion model in psychology literature (Ekman, 1992, 1999, 2016).

## 4.3 Training BERTsynth

We train BERTsynth, a BERT_base-cased model fine-tuned for synthetic text detection (using the *NEWSsynth* or RealNews-Test dataset). Input sequence length was maintained at the BERT maximum of 512 tokens ($\approx 384$ words). Five training runs were conducted. Each training run was 4 epochs – the most possible within GPU time constraints and similar to those of Zellers et al. (2019)

| GoodNewsEveryone | | Ekman |
| --- | --- | --- |
| disgust | → | disgust (8%) |
| fear | → | fear (8%) |
| sadness, guilt, shame | → | sadness (14%) |
| joy, trust, pride, love/like, positive anticipation/optimism | → | happiness (17%) |
| anger, annoyance, negative anticipation/pessimism | → | anger (24%) |
| negative surprise, positive surprise | → | surprise (30%) |

Table 1: Emotion Mapping Schema: GoodNewsEveryone (15 emotions) to Ekman 6 basic emotions. % shows the emotion label distribution in the dataset.

who used 5 epochs.[5] For each training run, a unique seed was used for model initialization, and a unique set of three seeds were used for the dataset shuffle - one seed each for train, validation, and test splits. Furthermore, the HuggingFace library shuffles the training data between epochs. The reproducibility of the training and validation results using seeds was verified by conducting multiple runs of training and validation. Hyperparameter values are listed in Appendix C.

### 4.4 Training emoBERT

We train emoBERT, a $BERT_{base}$-cased model fine-tuned on the single label multiclass task of emotion classification using the GoodNewsEveryone dataset. Fine-tuning emoBERT followed a similar process to fine-tuning BERTsynth described in §4.3. This time, there were 5k examples and fine-tuning was for 10 epochs.

Classification accuracy is not the end goal for emoBERT. Its purpose is to reduce the affective deficit of the PLM by modifying the representations of words conveying emotions and to improve performance in the task of synthetic text detection by transfer learning. The mean $F1_\mu$ for emoBERT is 39.4% on the Validation set - more than double mean chance (16.7%) and within the range 31% to 98% reported for within-corpus emotion classification in UnifiedEmotion (Bostan and Klinger, 2018). See Appendix D for more details.

---

### 4.5 Training emoBERTsynth

We train emoBERTsynth, an emoBERT model fine-tuned for synthetic text detection (using the *NEWSsynth* or RealNews-Test dataset). The best emoBERT model (checkpoint) from each of the 5 training runs had its emotion classification head (6 outputs) replaced with a binary classification head (2 outputs) for human vs synthetic text classification, see Figure 1. Each model was then fine-tuned on the synthetic text detection task using the exact same process and set of random seeds (for dataset shuffling) as the 5 best models described in §4.3. This allowed a direct comparison between the 5 BERTsynth models (trained on synthetic text detection only) and the 5 emoBERTsynth models (fine-tuned on emotion classification followed by synthetic text detection).

### 4.6 Results

The results in Figure 2 and Table 2 show the performance of BERTsynth and emoBERTsynth when fine-tuned on the *NEWSsynth* dataset. The results support the hypothesis that emotion can help detect synthetic text. emoBERTsynth outperforms BERTsynth in head-to-head for accuracy and F1 in all 5 runs.

Looking at precision and recall, emoBERTsynth outperforms BERTsynth in precision in all 5 runs, while the opposite is the case for recall. It is worth

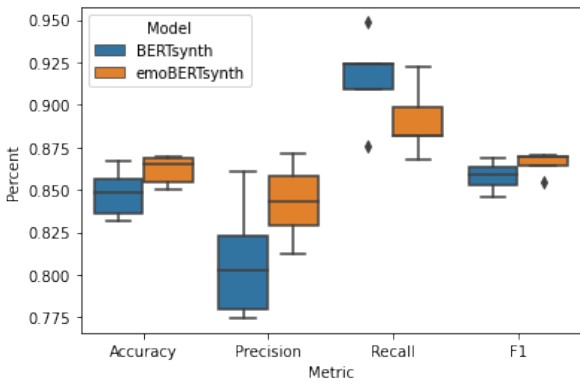

Figure 2: Test results for BERTsynth and emoBERTsynth on the *NEWSsynth* dataset. emoBERTsynth is higher for Accuracy, Precision and F1, while BERTsynth is higher for Recall.

comparing the relative difference in recall and precision between emoBERTsynth and BERTsynth models in Table 2. emoBERTsynth has a difference between the mean recall and mean precision of 4.76 (89.04 - 84.28) while the difference for BERTsynth is more than double that at 10.81 (91.63 - 80.82).

| | Precision | | Recall | | F1 | | Accuracy | |
|---|---|---|---|---|---|---|---|---|
| Run | Bs | emoBs | Bs | emoBs | Bs | emoBs | Bs | emoBs |
| 1 | 80.30 | 81.25 | 92.40 | 92.20 | 85.92 | 86.38 | 84.86 | 85.46 |
| 2 | 82.26 | 84.30 | 90.90 | 89.83 | 86.37 | 89.77 | 85.65 | 86.55 |
| 3 | 78.01 | 82.88 | 92.40 | 88.20 | 84.60 | 85.45 | 83.18 | 84.99 |
| 4 | 77.44 | 85.84 | 94.85 | 88.20 | 85.27 | 87.00 | 83.61 | 86.83 |
| 5 | 86.09 | 87.14 | 87.58 | 86.75 | 86.83 | 86.95 | 86.71 | 86.98 |
| Mean | 80.82 | **84.28** | **91.63** | 89.04 | 85.80 | **87.11** | 84.80 | **86.16** |
| Var. | (9.89) | (4.35) | (5.70) | (3.45) | (0.62) | (2.08) | (1.68) | (0.63) |
| $\Delta$ | +3.46 | | -2.59 | | +1.31 | | +1.36 | |

Table 2: Comparison of BERTsynth (Bs) and emoBERTsynth (emoBs) against the *NEWSsynth* test set. (Variance is shown in brackets under the mean). emoBs outperforms Bs in head-to-head for all 5 runs in Accuracy, F1, and Precision; while Bs outperforms emoBs in head-to-head for all 5 runs in Recall.

Thus, we suggest our emotionally-aware PLM, emoBERTsynth, is a better performing model than the standard PLM, BERTsynth, because it has a better balance between precision and recall.

In Table 3 we compare BERTsynth and emoBERTsynth on the RealNews-Test dataset. Recall that this dataset contains synthetic articles generated by Grover$_{mega}$ instead of the smaller Grover$_{base}$. We also compare against the FastText, GPT-2 and BERT detector models reported by Zellers et al. (2019) on this dataset. emoBERTsynth has the highest accuracy, outperforming BERTsynth by 1.4%, BERT$_{base}$ by 9.03%, GPT-2$_{base}$ by 10.03%, and FastText by 12.43%. These results support the hypothesis that emotion can improve synthetic text detection.

There is a 7.63 point difference between our BERTsynth model and the BERT model reported by Zellers et al. (2019), despite both models being BERT$_{base}$ and fine-tuned on the same dataset and splits. However, there are differences in how the models were treated before this fine-tuning, and there may be some hyperparameter differences for fine-tuning. We described in §4.3 how we fine-tune a randomly initialised BERT model to create BERTsynth. Zellers et al. (2019) reported their BERT models were domain adapted to News (by training on RealNews) at a length of 1024 WordPiece tokens. It is possible that this additional domain-adaptation and extended input sequence length actually harmed the performance of the BERT$_{base}$ model on the synthetic detection task. The performance of synthetic text detectors can improve with length (Ippolito et al., 2020) and the longer input sequence length could help in this regard. However, the vast majority of human and synthetic news articles in RealNews-Test are shorter than 1024 tokens. Thus, they may not benefit from that extended input length and the model may in fact be somewhat reliant on those later input tokens for prediction.

| Size | Model | Acc. |
|---|---|---|
| 11M | FastText | 63.80 |
| 124M | GPT-2$_{base}$ | 66.20 |
| | BERT$_{base}$ | 67.20 |
| | BERTsynth | 74.83 |
| | emoBERTsynth | **76.23** |

Table 3: emoBERTsynth outperforms other model architectures and sizes detecting human and Grover$_{mega}$ (1.5B) synthetic text from the RealNews-Test dataset. Detector model sizes include 11M and 124M parameters and architectures include FastText, GPT-2$_{base}$, and BERT$_{base}$. The FastText, GPT-2$_{base}$ and BERT$_{base}$ results are reported by Zellers et al. (2019).

## 4.7 Analysis

In this section, we perform a further set of experiments to aid in interpreting our main results.

### 4.7.1 Length of Human vs Synthetic articles

We investigate whether PLMs simply learn something about the length of articles as a proxy for discrimination between human and synthetic text. An analysis of NEWSsynth articles (train and validation splits) reveals no obvious correlation (Pearson $r = 0.20$) between the number of words in a human article and the resulting synthetic article. 64% of human articles are longer than their corresponding synthetic article, while 34% of synthetic articles

are longer. Human articles are longer overall, but have slightly shorter sentences than synthetic text; and human articles have more sentences per article - which accounts for their longer mean length. Similar observations were made for RealNews-Test by Bhat and Parthasarathy (2020). See Table 10 and Figs. 5 to 8 in Appendix E. Overall, these results point neither to article length nor sentence length as a reliable discriminator for synthetic text suggesting that detector models are not simply learning length as a proxy for human vs synthetic text.

### 4.7.2  Size of fine-tuning splits

| Split | Prec. | Recall | F1 | Acc. |
|---|---|---|---|---|
| 5-1-4k | 78.39 | 79.85 | 78.89 | 78.58 |
| Var. | (24.10) | (17.33) | (3.17) | (6.51) |
| 10-2-8k | 80.82 | 91.63 | 85.80 | 84.80 |
| Var. | (9.89) | (5.70) | (0.62) | (1.68) |
| $\Delta$ | +2.43 | +11.78 | +6.91 | +6.22 |

Table 4: BERTsynth metrics for different split sizes, using the *NEWSsynth* dataset averaged over 5 runs (with variance shown in brackets).

The BERTsynth fine-tuning regime (§4.3) was repeated using all (20k) and half (10k) of *NEWSsynth*. In all 5 runs, the BERTsynth model trained on the larger 20k dataset performed better than the equivalent model trained on the smaller 10k dataset – see Table 4. There was a modest improvement in precision (+2.43%) with a much larger increase in recall (+11.78%). The results suggest that recall is most sensitive to the size of the training set. This is perhaps because the PLM is already trained on human text during pretraining but not synthetic text (*exposure bias*), so more exposure to synthetic text increases the model's ability to detect synthetic text correctly with fewer false negatives.

### 4.7.3  Alternative forms of emoBERT

What is the effect of using different emotion datasets to fine-tune our emotionally aware PLMs on the downstream task of synthetic text detection? We conduct experiments on emoBERTsynth by fine-tuning eight alternative emoBERT models:

- **GNE** involves fine-tuning using the Good-NewsEveryone dataset (§4.2) as in the main experiments;
- **GNE$_r$** involves fine-tuning with a version of GNE with randomised labels. We do this to examine the extent to which the difference

between BERTsynth and emoBERTsynth can be attributed to emotion or to the process of fine-tuning on an arbitrary classification task with the GNE data;
- **AT** involves fine-tuning with the AffectiveText dataset comprising 1.5k news headlines in English annotated with respect to Ekman's 6 emotions (Strapparava and Mihalcea, 2008);
- **GA** is GNE and AT combined;
- **SST-2** involves fine-tuning on the task of sentiment polarity classification using the SST-2 dataset of 68,221 movie reviews in English (Socher et al., 2013);
- **GAS** is GNE, AT, and SST-2 combined; with SST-2 positive sentiment mapped to joy and negative sentiment mapped to sadness;
- **S-GA** involves first fine-tuning on sentiment using SST-2 and then fine-tuning on emotion using GA. This experiment is inspired by Kratzwald et al. (2018) who report that emotion classification can be improved by transfer learning from sentiment analysis;
- **GAS+-** is GAS but mapped to positive and negative sentiment.[6]

The results (Table 5) reveal that the best-performing emoBERTsynth models are those fine-tuned using GNE or using GNE and AffectiveText combined (GA). The latter achieves the highest accuracy and the former the highest F1. We attribute the relatively poor performance of AffectiveText on its own to its small size, comprising only 1.5k headlines (split 625 + 125 for training and dev splits respectively) compared to 5k for GNE and 68k for SST-2.

Table 5 also shows that fine-tuning on GNE outperforms fine-tuning with randomised labels (GNE$_r$). The 1.1 point drop in accuracy of GNE$_r$ compared to GNE suggests that the emotion classification task does play a role in the improved performance of emoBERTsynth versus BERTsynth.

The results in Table 5 suggest that fine-tuning on sentiment is not particularly helpful. The poor performance of GAS could be due to the crude mapping of negative sentiment to sadness (because

---

[6]Happiness was mapped to positive sentiment; sadness, fear, anger and disgust were mapped to negative sentiment; surprise was mapped to sentiment using a DistilBERT (base-uncased) (Sanh et al., 2020) sentiment classifier fine-tuned on the SST-2 dataset and available on HuggingFace. https://huggingface.co/distilbert-base-uncased 14.05% of 'surprise' mapped to positive, while the remaining 85.95% mapped to negative sentiment.

|        | Prec. | Rec. | F1 | Acc. |
|--------|-------|------|------|------|
| GAS    | 81.95 | 85.58 | 83.72 | 83.36 |
| S-GA   | 82.60 | 87.80 | 85.12 | 84.65 |
| GAS+-  | 82.41 | 88.30 | 85.25 | 84.73 |
| AT     | **85.52** | 83.88 | 84.69 | 84.84 |
| SST-2  | 82.85 | 88.38 | 85.52 | 85.04 |
| GNEr   | 82.44 | 89.93 | 86.02 | 85.39 |
| GNE    | 83.84 | **90.40** | **87.00** | 86.49 |
| GA     | 85.34 | 88.18 | 86.73 | **86.51** |

Table 5: Ablation experiments, using different emotion datasets for fine-tuning emoBERT, comparing emoBERTsynth (eBs) detectors on the task of synthetic text detection on the *NEWSsynth* dataset. GNE is the GoodNewsEveryone dataset which is used in the main experiments. GNE$_r$ is GNE with randomised labels. AT is AffectiveText. GA is GNE and AT combined. SST-2 is the SST-2 sentiment dataset. GAS is the combined GNE, AT, and SST-2 datasets. S-GA is first fine-tuned on sentiment using the SST-2 dataset, and then fine-tuned on emotion using the GNE and AT datasets, and finally fine-tuned on synthetic text detection. GAS+- is GAS but mapped to positive and negative sentiment.

it could be any 1 of 5 Ekman emotions), which results in a large dataset imbalance across emotion labels. When we go in the opposite direction and mapped the emotion labels to sentiment labels (GAS+-), the results improved. Overall, however, the results suggest that mixing emotion and sentiment datasets is not a good idea (particularly if they are disproportionate in size and imbalanced), and that sentiment alone is not sufficient.

#### 4.7.4 A larger detector model

We next investigate what happens when we use a PLM larger than BERT to detect synthetic text. Using the same experimental setup described in §4, we substituted BLOOM (Scao et al., 2023) in place of BERT for the synthetic text detector. BLOOM is an open-science causal PLM alternative to GPT-3 (Brown et al., 2020). We use the BLOOM 560M size model. The results in Table 6 show that the emotionally-aware BLOOM PLM (emoBLOOMsynth) outperforms the standard BLOOM (BLOOMsynth) in all metrics.

## 5 ChatGPT Experiments

All experiments so far have involved PLMs pretrained with the self-supervised objective of predicting the next token or a masked token. We conduct a final experiment with ChatGPT, a more human-

|               | Prec. | Rec. | F1 | Acc. |
|---------------|-------|------|------|------|
| BLOOMsynth    | 81.90 | 85.95 | 83.79 | 83.40 |
| Var.          | (4.76) | (12.22) | (1.23) | (0.93) |
| emoBLOOMsynth | **85.98** | **88.02** | **86.90** | **86.75** |
| Var.          | (5.72) | (9.96) | (0.27) | (0.15) |
| Δ             | +4.08 | +2.07 | +3.11 | +3.35 |

Table 6: Comparison of BLOOMsynth and emoBLOOMsynth against the *NEWSsynth* test set averaged over 5 runs (with variance in brackets). emoBLOOMsynth outperforms BLOOMsynth in Accuracy, F1, Recall, and Precision.

aligned Large Language Model (LLM) which has undergone a second training or "alignment" phase using Reinforcement Learning from Human Feedback on top of an underlying LLM (GPT 3.5 in our case) (OpenAI, 2022; Ouyang et al., 2022). We create a custom dataset comprising human articles and ChatGPT synthetic text from multiple non-news domains, and use it to compare our BERTsynth and emoBERTsynth models against ChatGPT (in a zeroshot setting) on the task of detecting ChatGPT's own synthetic text.[7]

**ChatGPT100** We create and release *ChatGPT100* - a dataset comprising human articles and synthetic articles generated by ChatGPT. Following Clark et al. (2021) who collected 50 human articles and generated 50 articles using GPT2 and GPT3, we also collect 50 human articles, and we then use ChatGPT to generate 50 synthetic ones. The human written articles are from 5 different domains: Science, Entertainment, Sport, Business, and Philosophy. We used reputable websites for the human text which was gathered manually, see Table 8 in Appendix B.3. The synthetic text was generated by providing ChatGPT with a prompt such as "*In less than 400 words, tell me about moral philosophy.*" where human text on the same topic, moral philosophy in this case, had already been found online. The data generated by ChatGPT is semantically correct and was checked manually. Subject areas in which the authors are knowledgeable were chosen so that the correctness of the synthetic text could be checked. To be comparable with the detectors presented in our earlier experiments, the articles were limited to a maximum of 384 words ($\approx$ 512 tokens) and truncated at a natural sentence boundary. The two articles were then made to be approximately the same length.

---

[7]We use ChatGPT-3.5 (Mar-14-2023 version) between dates 16-Mar-2023 and 24-Mar-2023.

| Model | Prec. | Rec. | F1 | Acc. |
|---|---|---|---|---|
| ChatGPT | **75.00** | 30.00 | 42.86 | 60.00 |
| BERTsynth | 60.24 | **100.00** | 75.19 | 67.00 |
| emoBERTsynth | 67.57 | **100.00** | **80.65** | **76.00** |

Table 7: Our emotionally aware PLM (emoBERTsynth) outperforms ChatGPT and BERTsynth at detecting synthetic text in the *ChatGPT100* dataset. Note that Chat-GPT is performing the task zero-shot.

**Detection task** Each article was appended to the following prompt to ChatGPT: "*Was the following written by a human or a computer, choose human or computer only?*" Having tested ChatGPT, we then tested our BERTsynth and emoBERTsynth models (the models fine-tuned on RealNews-Test from Table 3).

**Results** The results are shown in Table 7. The first thing to note is that no model performs particularly well. ChatGPT tends to misclassify its own synthetic text as human (hence the low recall score of 30%).[8] BERTsynth and emoBERTsynth, on the other hand, tend to classify text as machine-written and they both obtain 100% recall. We previously saw (§4.7.2) that recall is most sensitive to fine-tuning set size. The emoBERTsynth and emoBERTsynth models have been exposed to synthetic text during fine-tuning, whereas ChatGPT is performing the task zero-shot. This could explain some of the difference in recall between the ChatGPT and the two fine-tuned models.

Finally, as with our experiments with Grover-generated text, emoBERTsynth outperforms BERTsynth on all metrics. The dataset is small so we must be careful not to conclude too much from this result, but it does suggest that fine-tuning on emotion could be beneficial when detecting synthetic text from LLMs and more sophisticated generators, in non-news domains. This is in line with the results of our earlier experiments using variously size PLMs (such as Grover, BERT, BLOOM), used as generators and detectors in the news domain, and shows the potential for our approach with different generator models and in different domains.

## 6 Conclusion

We conducted experiments investigating the role that emotion recognition can play in the detection

---

[8]ChatGPTs responses suggest it may use fact-checking as a proxy during synthetic text detection.

of synthetic text. An emotionally-aware PLM fine-tuned on emotion classification and subsequently trained on synthetic text detection (emoPLMsynth) outperformed a model with identical fine-tuning on synthetic text detection, but without emotion training, (PLMsynth). The results hold across different synthetic text generators, model sizes, datasets and domains. This work specifically demonstrates the benefits of considering emotion in the task of detecting synthetic text, it contributes two new datasets (*NEWSsynth* and *ChatGPT100*) and, more generally, it hints at the potential benefits of considering human factors in NLP and Machine Learning.

Is it possible that some other proxy for synthetic text is at play? We ruled out some potential proxies related to article length in §4.7.1. In ablation studies in §4.7.3, we showed that the emotion labels result in an improvement in performance compared to randomized labels for the same emotion dataset. Other potential proxies are nonsensical sentences, repetitive text, etc. However, we account for these by comparing our emotionally-aware PLMs (emoPLMsynth) against standard PLMs fine-tuned on synthetic text detection only (PLMsynth). Thus, any advantage or disadvantage of sentences without meaning (or any other factor) is also available to the non-emotionally-aware model against which we compare our emotionally-aware model.

Future work will investigate further the *affective profile* (i.e. emotional content and characteristics) of human and synthetic text; and attempt to determine if there are measurable differences which may prove useful in the task of synthetic text detection.

## Limitations

The datasets used in this work (synthetic text datasets, emotion datasets, and sentiment dataset) are English language and model performance in other languages may vary. We primarily focus on the news domain and, while performance in other domains may vary (Merchant et al., 2020), we include experiments in several non-news domains (§5).

The emotion datasets are imbalanced across emotion labels which can impact overall performance, and we conducted ablation experiments to find the best combination of emotion and sentiment datasets (§4.7.3). GoodNewsEveryone's 15 emotions were mapped to Ekman's 6 emotions (Ekman, 1992, 1999, 2016), factoring in Plutchik's wheel of emotion (Plutchik, 1980, 2001), but there is no

firm agreement in the literature as to which is the 'correct' or 'best' emotion model (Ekman, 2016). The emotion models used in this work are the two most popular in the literature.

The maximum input sequence length of BERT is 512 tokens and articles longer than this are truncated, which may negatively affect performance on the synthetic text detection task (Ippolito et al., 2020). However, we also saw that increasing the input sequence length may actually contribute to poorer performance (§4.6).

## Ethical Considerations

We release multiple PLMs (emoBERTsynth, BERTsynth, emoBLOOMsynth and BLOOMsynth) which we refer to generically as emoPLMsynth and PLMsynth. emoPLMsynth and PLMsynth are BERT or BLOOM models with versions fine-tuned on *NEWSsynth* or the RealNews-Test (Zellers et al., 2019) datasets; emoPLMsynth is also fine-tuned on combinations of the GoodNewsEveryone (Bostan et al., 2020), AffectiveText (Strapparava and Mihalcea, 2008), and SST-2 (Socher et al., 2013) datasets.

We release *ChatGPT100*, a dataset comprising 100 English language articles in various non-news domains. 50 articles are human written, and 50 articles are generated by ChatGPT. The 100 articles have all been manually curated and do not contain toxic content. Furthermore, ChatGPT has a content filter which flags potentially harmful content.

We release, *NEWSsynth*, a dataset comprising 40k English language articles in the news domain. [9] 20k news articles are human (from RealNews-Test) and 20k generated by Grover. Publishing synthetic text is a risk, but *NEWSsynth* is clearly labelled as containing synthetic text. This is a similar precaution to synthetic text from Grover which has already been published and is publicly available (Zellers et al., 2019).

The potential harms, such as toxic synthetic text (Gehman et al., 2020), of PLMs pretrained on web-crawled data has been the subject of much discussion (Bender et al., 2021). Since emoPLMsynth and PLMsynth (and Grover) were pretrained and/or fine-tuned on web-crawled data there is a possibility they could produce inappropriate synthetic text and this includes the *NEWSsynth* dataset. We recognise these potential harms and to mitigate them

___
[9]We include 20k articles in addition to the 20k used in this work

include the caveat below with the released datasets (*NEWSsynth* and *ChatGPT100*) and the released language models (emoPLMsynth, PLMsynth):

> Care must be taken when using these language models (emoPLMsynth and PLMsynth), and datasets (*NEWSsynth* and *ChatGPT100*) as they may produce or contain ethically problematic content. Data scraped from the web may contain content which is ethically problematic such as adult content, bias, toxicity etc. and web-scraped data is used in the pre-trained language models such as BERT, BLOOM and Grover. PLMsynth and emoPLMsynth are based on BERT or BLOOM PLMs, while *NEWSsynth* was generated by Grover. Consequently, emoPLMsynth and PLMsynth could produce text which is ethically problematic, while *NEWSsynth* may contain ethically problematic content. As a result, any use of the language models (emoPLMsynth, PLMsynth) or the datasets (*NEWSsynth* or *ChatGPT100*) should employ appropriate checks and test regimes to handle potential harmful content.

The intended use of the emoPLMsynth and PLMsynth models, and the *NEWSsynth* and *ChatGPT100* datasets, is for research purposes and beneficial downstream tasks such as identifying synthetic text perhaps in online news, reviews, comments, plagiarism etc. Online platforms could use this identification to decide whether or not to publish such content, or where to surface it via recommender algorithms etc. This could help protect public confidence in online discourse.

Energy usage was reduced by training on smaller models and for a relatively small number of epochs where possible, by using random search rather than an exhaustive grid search, and by using freely available managed compute resources where possible.

## Acknowledgements

This publication has emanated from research conducted with the financial support of Science Foundation Ireland under Grant number 18/CRT/6183. For the purpose of Open Access, the author has applied a CC BY public copyright licence to any Author Accepted Manuscript version arising from this submission.

We thank Rowan Zellers for his permission to use 20k human articles from RealNews-Test (Zellers et al., 2019) in the *NEWSsynth* dataset which we release with this paper.

We thank the reviewers for their valuable feedback which helped improve the paper.

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

## A    Plutchik Wheel of Emotion

The Plutchik Wheel of Emotion (Plutchik, 1980, 2001) is shown in Figure 3 and is the most commonly used dimensional model in psychology literature.

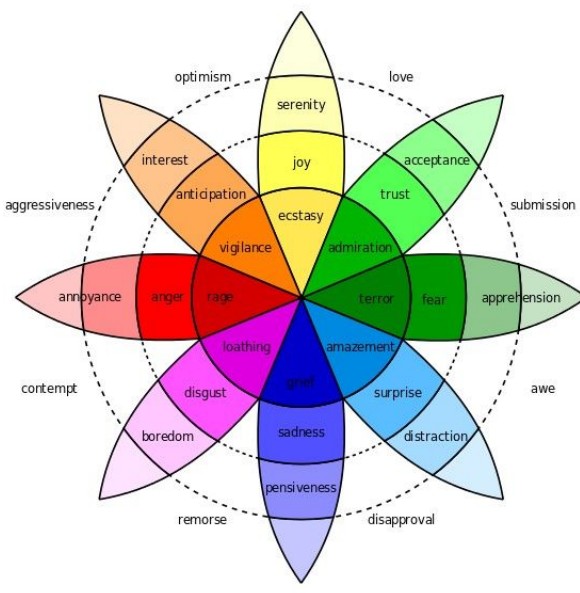

Figure 3: Plutchik Wheel of Emotion. The middle ring includes Ekman's 6 emotions plus 'trust' & 'anticipation'. Similar emotions lie on adjacent spokes e.g. anger-disgust, while opposing emotions are placed on opposing spokes e.g. joy-sadness.

## B    Reproducibility

All code, models, and datasets (including *NEWSsynth* and *ChatGPT100*) are available at https://github.com/alanagiasi/emoPLMsynth.

### B.1    Parameters used for generating synthetic text with Grover

Grover$_{BASE}$ was used for generating synthetic text news articles in *NEWSsynth*. Full contextual metadata was used, in addition to a top-p value of 0.95 because both can make discrimination more difficult. According to Zellers et al. (2019) contextual data decreased perplexity by 0.9 points for Grover$_{BASE}$, and a top-p value in the range 0.92 to 0.98 is a Goldilocks zone where discrimination is

hardest (so we chose top-p=0.95 as it is in the middle of this difficult detection zone). Source code, installation, and generation instructions for Grover can be found on the Grover github. [10]

- Model: GROVER$_{BASE}$
- Model parameters: 124M
- Top-p = 0.95
- Metadata: Full contextual metadata (from RealNews-Test dataset)
- Time to generate 20k synthetic articles is approximately 90 hours on a single GPU (Tesla K40, or RTX2080ti) with 30GB RAM.

### B.2    Metrics

Accuracy, Precision, Recall, F1, (and F1$_\mu$ for emotion classification) were calculated using scikit-learn. [11]

### B.3    Datasets

**NEWSsynth**    We release *NEWSsynth* - a dataset comprising 40k English language human and synthetic news articles. The experiments in this paper use the first 20k of these articles, an additional 20k articles are provided in the dataset. The human articles are taken from the RealNews-Test dataset (Zellers et al., 2019) so they have not been seen by Grover - which generated the synthetic news articles as described in §4.3 and earlier in this Appendix.

Regarding the emotional content and journalistic content of news articles in *NEWSsynth*: Previous authors have specifically chosen the news domain because of its high emotional content (Strapparava and Mihalcea, 2007; Bostan et al., 2020). It is long established that different emotions lead to different actions (Spielberger, 1972) including what we write (Brand, 1985). Emotion can be exploited, for example "engagement based ranking" tends to favour content that evokes anger (Haugen, 2021). While some journalistic reporting is objective, opinion editorials (op-ed) are opinions pushing an agenda and, for example, tabloids tend

---

[10]https://github.com/rowanz/grover
[11]scikit-learn describes the metrics: https://scikit-learn.org/stable/modules/model_evaluation.html#common-cases-predefined-values. As noted in §4.4 when fine-tuning emoBERT on emotions: micro averaging over a single-label multi-class evaluation means that Accuracy, Precision, Recall and F1 all have the same value. https://scikit-learn.org/stable/modules/model_evaluation.html#multiclass-and-multilabel-classification

to specifically exploit emotion. The 10k news articles in the *NEWSsynth* training split, for example, come from 150 online sources which also include: movie reviews and entertainment such as rollingstone.com, hollywoodlife.com, bollywood-hungama.com and mashable.com; and tabloids such as thedailymail.co.uk, dailystar.co.uk, thedailystar.net etc. which cover many types of news including journalism, op-eds, reviews, opinions etc. In short, *NEWSsynth* is not limited to non-emotional objective fact reporting, it contains a broad spectrum of journalistic styles and content.

**ChatGPT100**   We release *ChatGPT100*, a dataset comprising 100 English language articles in various non-news domains (Science, Entertainment (Music, Movies), Sport, Business, and Philosophy). 50 articles are human written, and 50 articles are generated by ChatGPT. The 100 articles have all been manually curated and do not contain toxic content. Furthermore, ChatGPT has a content filter which flags potentially harmful content.

The 50 human articles contained in *ChatGPT100* were gathered between 16-24 March 2023 from the domains shown in Table 8. The 50 synthetic articles contained in *ChatGPT100* were generated using ChatGPT 3.5 (March 14 2023 version) on dates between 16-24 March 2023.

**RealNews and RealNews-Test**   These datasets were released with Grover and are described there in detail (Zellers et al., 2019).

**Emotion and Sentiment Datasets**   GoodNewsEveryone is described in detail (Bostan et al., 2020) with modifications made to the dataset for this work described in §4.2. The distribution of emotion intensity is shown in Table 9 showing almost all are 'medium' while 2 examples have no emotion. AffectiveText was released as part of SemEval 2008 and is described in detail (Strapparava and Mihalcea, 2008), while the SST-2 sentiment dataset is described in detail (Socher et al., 2013).

## C   Hyperparameters used for Fine-tuning

The hyperparameters used for PLM fine-tuning are listed below. If not specifically listed, the hyperparameter value used was the default using HuggingFace Transformer libraries. [12] The BERT$_{BASE}$-cased and BERT$_{LARGE}$-cased models were down-

| Domain | Count |
|---|---|
| britannica.com | 9 |
| investopedia.com | 6 |
| plato.stanford.edu | 6 |
| fandom.com | 2 |
| forbes.com | 2 |
| olympics.com | 2 |
| allmusic.com | 1 |
| arpansa.gov.au | 1 |
| arsenal.com | 1 |
| atptour.com | 1 |
| bbc.com | 1 |
| bhf.org.uk | 1 |
| bleacherreport.com | 1 |
| cambridge.org | 1 |
| canarahsbclife.com | 1 |
| empireonline.com | 1 |
| gaa.ie | 1 |
| hotpress.com | 1 |
| kaspersky.com | 1 |
| laureus.com | 1 |
| mayfieldclinic.com | 1 |
| oah.org | 1 |
| oceanservice.noaa.gov | 1 |
| open.lib.umn.edu | 1 |
| phys.org | 1 |
| science.nasa.gov | 1 |
| sixnationsrugby.com | 1 |
| slf.rocks | 1 |
| u2.com | 1 |
| Total: | 50 |

Table 8: Domains used for human text in *ChatGPT100* dataset released with this paper. Articles were gathered between 16-24 March 2023.

loaded from HuggingFace. [13]

`emoBERT`, `BERTsynth`, and `emoBERTsynth` were all trained using freely available Google Colab with a single GPU (Tesla K80 or Tesla T4) with no guarantee on available RAM [14] or an NVIDIA GeForce RTX3090 GPU with 24GB RAM.

All models were trained and evaluated for 5 runs using different seeds for each of the 5 runs. The seeds used are listed below.

---

[12]https://huggingface.co/transformers/

[13]https://huggingface.co/bert-base-cased, https://huggingface.co/bert-large-cased

[14]GPU time was typically limited to 6 hours or less which limited the number of epochs the PLMs could be trained to 4.

| Emotion Intensity | Count |
|---|---|
| None | 2 |
| low | 3 |
| medium | 4991 |
| high | 4 |
| Total: | 5000 |

Table 9: Distribution of emotion intensity in GoodNewsEveryone.

## C.1 BERTsynth, emoBERTsynth

- Model: BERT$_{BASE}$-cased | BERT$_{LARGE}$-cased
- Model parameters: 110M | 355M
- Input sequence length: 512 tokens padded
- Train-Val-Test split size: 10k, 2k, 8k
- Epochs: 4 | 5
- Batch Size: 7. [15]
- Batch Gradient Accumulation: 8
- Warmup steps: 500
- Weight decay: 0.01
- Seeds = [179, 50, 124, 253, 86]. 5 seeds = 1 seed per training run.
- Data seeds = [17, 38, 5, 91, 59] #n, n-6, n+6 for train-val-test seeds respectively
- Metric for best model: Accuracy
- Training + Validation time: 150mins (for 4 epochs)
- Inference time: 10mins (for 8k examples)

## C.2 emoBERT

- Model: BERT$_{BASE}$-cased | BERT$_{LARGE}$-cased
- Model parameters: 110M | 355M
- Input sequence length: 512 tokens padded
- Train-Val-Test split size: 2.5k, 0.5k, 2k for GNE
- Epochs: 10
- Batch Size: 7
- Batch Gradient Accumulation: 8
- Warmup steps: 500
- Weight decay: 0.01
- Seeds = [179, 50, 124, 253, 86]. 5 seeds = 1 seed per training run.

---

[15]Experiments showed 7 was the largest batch size possible given a 512 input sequence length with the RAM available, and is similar to that reported by Google on BERT github: https://github.com/google-research/bert

- Data seeds = [17, 38, 5, 91, 59] #n, n-6, n+6 for train-val-test seeds respectively
- Metric for best model: F1$_\mu$
- Training + Validation time: 11mins (for 10 epochs)
- Inference time: 22s (for 2k examples)

## D emoBERT Emotion Classification Results

The mean F1$_\mu$ for emoBERT is 39.4% on the Validation set - more than double mean chance (16.7%) and within the range 31% to 98% (mean = 62.6%) reported for within-corpus emotion classification in UnifiedEmotion (Bostan and Klinger, 2018). GoodNewsEveryone does not report news headline emotion classification (Bostan et al., 2020).

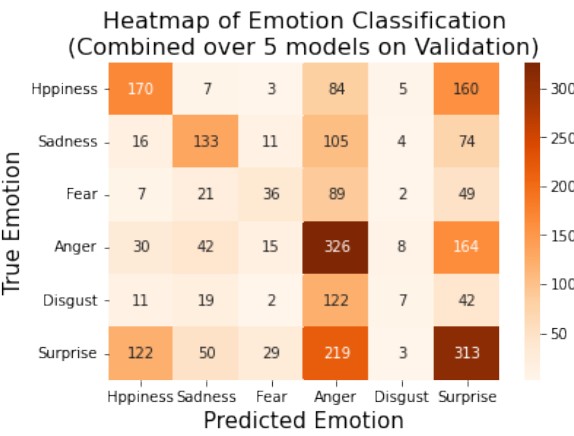

Figure 4: Combined Confusion Matrix for Emotion Classification on GoodNewsEveryone Validation set.

Figure 4 depicts the combined results of the best performing model (in Validation) from the 10 epochs, in each of the 5 training runs. The imbalance across emotion labels (shown in the first column of Table 1) is reflected in performance in Figure 4. Anger and Surprise are the two emotions best classified and best represented in the dataset at 24% and 30% respectively; while Fear and Disgust are the two emotions most poorly classified and least represented in the dataset at 8% each. The 4 emotions Happiness, Sadness, Anger, and Surprise are classified correctly more often than as any of the other 5 emotions. Fear and Disgust are most likely to be misclassified as Anger.

We see a correlation between class size and performance on that class - those classes with more examples performed better than those with fewer examples. Outright performance is not the end

goal for emoBERT. The purpose of emoBERT is to reduce the *affective deficit* of the PLM by modifying the word representations of words representing emotions and to improve performance in the task of synthetic text detection by transfer learning.

## E Length of human vs synthetic articles in NEWSsynth

Figures 5 - 8 illustrate the relative lengths of human and synthetic articles and sentences in *NEWSsynth* (train and validation splits) as described in §4.7.1 and shown in Table 10.

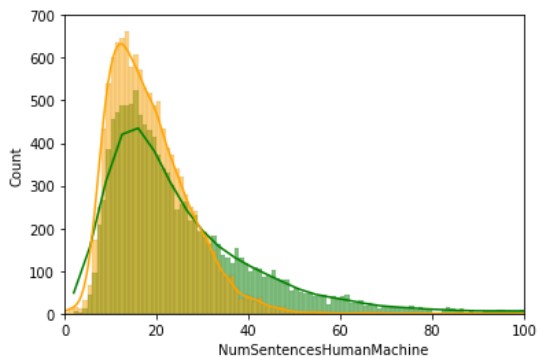

Figure 7: Number of sentences per article for human (green) and synthetic (orange) text in NEWSsynth.

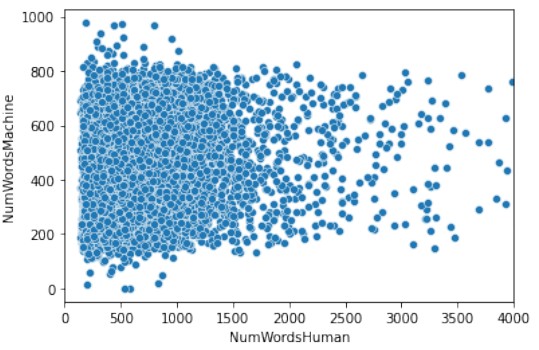

Figure 5: Scatter plot of number of words per article pair of synthetic text vs. human text in NEWSsynth (Pearson $r = 0.20$).

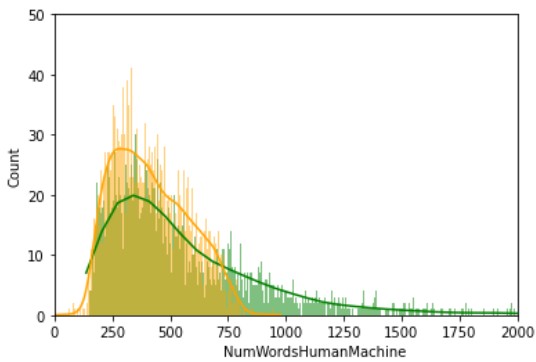

Figure 6: Number of words per article for human (green) and synthetic (orange) text in NEWSsynth.

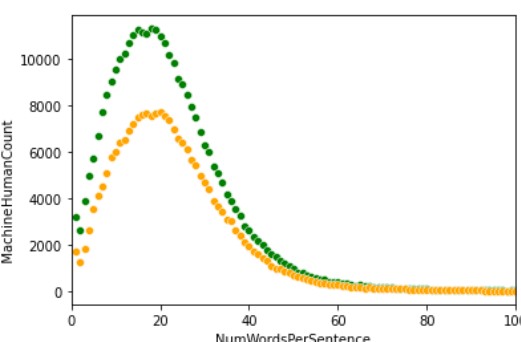

Figure 8: Number of words per sentence for human (green) and synthetic (orange) text in NEWSsynth.

| Source | Words Per Article | | Sentences Per Article | | Words Per Sentence | |
|---|---|---|---|---|---|---|
| | $\overline{x}$ | $\sigma$ | $\overline{x}$ | $\sigma$ | $\overline{x}$ | $\sigma$ |
| Human | 594.56 | 503.07 | 27.05 | 25.23 | 21.98 | 15.98 |
| Synthetic | 417.98 | 162.09 | 18.34 | 8.64 | 22.79 | 16.60 |
| | Figure 6 | | Figure 7 | | Figure 8 | |

Table 10: Comparison of Human and synthetic text in the *NEWSsynth* dataset showing the mean ($\overline{x}$) and standard deviation ($\sigma$) for Word Per Article, Sentences Per Article, and Words Per Sentence. Human articles are longer overall, but have slightly shorter sentences than synthetic text; and Human articles have more Sentences Per Article - which accounts for their longer mean length.