# OpenReview forum: "Do Stochastic Parrots have Feelings Too? Improving Neural Detection of Synthetic Text via Emotion Recognition"
_EMNLP/2023/Conference — EMNLP 2023 Findings_

### Official Review · Reviewer_DyWC · 2023-08-03

**Soundness:** 3

**Excitement:**

3: Ambivalent: It has merits (e.g., it reports state-of-the-art results, the idea is nice), but there are key weaknesses (e.g., it describes incremental work), and it can significantly benefit from another round of revision. However, I won't object to accepting it if my co-reviewers champion it.

**Paper Topic And Main Contributions:**

This paper studies the task of distinguishing synthetic texts from human-written texts. The key idea in the paper is that synthetic texts might lack emotional patterns that are present in human-written texts. Based on the idea, they developed a detection model that is aware of emotion by fine-tuning a detection model first on an emotion classification dataset and then the dataset of synthetic and human-written texts. They conducted experiments and showed that the additional fine-tuning on emotion classification improves the detection model.

**Reasons To Accept:**

* The idea is interesting that there is a lack of / there are different patterns of emotions in synthetic texts, which could be leveraged to aid detection of synthetic texts.

* The proposed method is simple. The main experiments show that fine-tuning on an emotion classification dataset could improve the detection performance.

* The authors also conducted several ablation experiments to help better understand the factors that affect the model's performance.

**Reasons To Reject:**

The main weakness is that there is not enough analysis to support its main claim - emotion-awareness could be useful in detecting synthetic texts. Here are the reasons:

*  The experiments are only conducted in the domain of news, where texts are usually more objective and there are less emotion expressions. So detecting emotion might help distinguish synthetic news and human-written news (as emotion is less frequent in this domain, the generation model is less likely to learn emotion during pretraining). The effects of emotion-awareness remain unclear for other domains such as movie reviews and blog posts. As emotion expressions are more frequent in these domains, synthetic generators pretrained on data in these domains might have already learned to generate texts with emotions. As a result, experiments in other domains are important to support the main claim in this work.

* Table 6 gives contradicting evidence. For example, AT fine-tunes the detection model with emotion classification data of news headlines, but it achieves a worse F1 score (84.69)  than BERTsynth (85.8 F1 in Table 2). This suggests that emotion-awareness might not be useful. Another example is GAS, which achieves the lowest F1 score of 83.72 in spite of the fact that it fine-tunes the model with 3 emotion classification datasets. As a result, some results do not support the claim that emotion-awareness could be useful for synthetic text detection.

* Although the paper conducts several experiments to understand the effects of different factors on the model performance, it still lacks some analysis of why using emotion data improves the detection performance. Is it because synthetic texts do not have much emotion, or is it because it contains inconsistent emotions? At least give some examples where the predictions were corrected after fine-tuning over the emotion data, so readers could better understand how emotion-awareness might help in these cases.

**Reproducibility:**

4: Could mostly reproduce the results, but there may be some variation because of sample variance or minor variations in their interpretation of the protocol or method.

**Reviewer Confidence:**

3: Pretty sure, but there's a chance I missed something. Although I have a good feel for this area in general, I did not carefully check the paper's details, e.g., the math, experimental design, or novelty.

**Typos Grammar Style And Presentation Improvements:**

There are some grammatical errors in this paper (e.g., row 5 - highlight). Please improve the writing.

---

> ### Author Rebuttal · Authors · 2023-08-29
>
> Thank you for the feedback and suggestions.
>
> **Reasons to Reject:**
> >“The experiments are only conducted in the domain of news …”
>
> We conducted experiments in 5 non-news domains as described in lines 531-534 and release them in the ChatGPT100 dataset with this work. We highlight the unique significance of the news domain in lines 045-048 “*Due to the potentially profound consequences of global synthetic disinformation we focus mainly, but not exclusively, on the detection of synthetic text in the news domain.*” Furthermore, our emotionally-aware models, fine-tuned in the news-domain, outperform ChatGPT at detecting non-news synthetic text - which supports the claim of this work beyond the news domain. See also our answer below regarding the varied news domains included in the NEWSsynth dataset which we release with this work.
>
> >“... in the domain of news, where texts are usually more objective and there are less emotion expressions …”.
>
> We disagree with this assertion. On the contrary, other authors have specifically chosen the news domain because of its high emotional content (Strapparava and Mihalcea, 2007) (Bostan et al., 2020). It is long established that different emotions lead to different actions (Spielberger, 1972) including what we write (Brand, 1985). Emotion can be exploited, for example “engagement based ranking” tends to favour content that evokes anger (Haugen, 2021).
> While some journalistic reporting is objective, opinion editorials (op-ed) are opinions pushing an agenda and, for example, tabloids tend to specifically exploit emotion. The 10k news articles in NEWSsynth training split come from 150 online sources including: “*movie reviews*” (which your review mentioned) and entertainment such as rollingstone.com, hollywoodlife.com, bollywoodhungama.com and mashable.com; and tabloids such as thedailymail.co.uk, dailystar.co.uk, thedailystar.net etc. which cover many types of news including journalism, op-eds, reviews, opinions etc.
>
> >“Table 6 gives contradicting evidence. …”
>
> Just to clarify, we believe it is Table 5 you are referring to. AT is a small dataset of 1.5k news headlines, of which 625 + 125 were used in the training and dev splits respectively. We attribute the poorer performance of AT to the dataset size being insufficiently large (rather than emotion itself underperforming).
> GAS includes the SST-2 movie review sentiment dataset which is orders of magnitude larger than the AT and GNE datasets. Table 5 shows that, apart from AT alone, any combination of datasets containing sentiment underperforms the models fine-tuned on emotion alone. For GAS, we mapped positive sentiment to joy and negative sentiment to sadness, which is crude because it could be 1 of 5 emotions, and which results in a dataset imbalance. (Apologies we did not make this clear in the paper but will do so in the revision). When we went in the opposite direction and mapped the emotion labels to sentiment labels (GAS+-),  the results improved. Overall, however, we conclude that mixing emotion and sentiment datasets is not a good idea, and that sentiment alone is not sufficient.
>
> >“...  Is it because synthetic texts do not have much emotion, or is it because it contains inconsistent emotions? …”
>
> This is an important point, we will update the paper with examples of emotion incongruence and/or lack of emotion. See examples in our response to Reviewer 8S6u, repeated below for convenience:
>
> Examples where emoLLMsynth detected synthetic text that BERTsynth didn’t and there is affective coherence.
> - E.g. 1 "Marshal Yanda looked "**happy** and refreshed" before Wednesday's minicamp practice but made some notes. After a **disappointing** season, the 37-year-old Yanda might be "just getting **better and better and better**" physically, veteran linebacker Whitney Mercilus said.”
> - E.g. 2 "Bethenny Frankel, who hosted her daughter, Brynn, on ‘Bethenny Ever After,’ opened up about how she found balance when dealing with **two recent heartbreaks**. Bethenny Frankel, 39, has had a **rocky relationship** with her child, Bryn, 11, but all the drama behind the scenes has her engaged in **peace and happiness**.”

---

### Official Review · Reviewer_TNw9 · 2023-08-03

**Typos Grammar Style And Presentation Improvements:** I suggest authors to read the paper c…
**Soundness:** 4

**Ethical Concerns:**

Yes

**Excitement:**

3: Ambivalent: It has merits (e.g., it reports state-of-the-art results, the idea is nice), but there are key weaknesses (e.g., it describes incremental work), and it can significantly benefit from another round of revision. However, I won't object to accepting it if my co-reviewers champion it.

**Justification For Ethical Concerns:**

Authors already stated an ethical discussion. Probably these claims should be properly avaluated.

**Missing References:**

None

**Paper Topic And Main Contributions:**

The authors propose to perform a fine-tuning phase of LLMs on the emotion detection task before performing an additional fine-tuning phase on data for synthetic text identification.

**Questions For The Authors:**

What is the prompt used for data generation?
Are data generated semantically correct?
LLMs models you sed are very small and sentences without meaning are often generated. Can your results be influenced by this issue?
Why do you not evaluate other simplest strategies for text classification as baseline?

**Reasons To Accept:**

- The paper il easy to read and understand
- The novelty is good enough
- Results seems encouraging

**Reasons To Reject:**

- No replicability of results
- No statistical validation of results are performed
- Only one dataset has been used
- More classic machine learning models should be used as baselines.

**Reproducibility:**

3: Could reproduce the results with some difficulty. The settings of parameters are underspecified or subjectively determined; the training/evaluation data are not widely available.

**Reviewer Confidence:**

3: Pretty sure, but there's a chance I missed something. Although I have a good feel for this area in general, I did not carefully check the paper's details, e.g., the math, experimental design, or novelty.

---

> ### Author Rebuttal · Authors · 2023-08-29
>
> Thank you for the feedback and suggestions.
>
> **Reasons to Reject:**
> >“No replicability of results”
>
> We are not sure what you mean by this? All of the experiments are reproducible. All hyperparameters etc. are included in Appendices B and C. All code, datasets, and models will be available publicly (Github, HuggingFace) outside of the anonymity period.
>
> >“Only one dataset has been used”
>
> We used several datasets: RealNews-Test, NEWSsynth, and ChatGPT100 were used for the synthetic text detection task (we release the NEWSsynth, and ChatGPT100 with this paper). GoodNewsEveryone, AffectiveText, and SST-2 were used for the emotion and sentiment tasks.
>
> >“No statistical validation of results are performed”
>
> We will include the statistical validation of results. However, Table 2 for example is averaged over 5 runs with all the results provided, so we are not sure it is necessary in such a case.
>
> >“More classic machine learning models should be used as baselines.”
>
> We take your point, however in this paper we use LLMs as they provide state-of-the-art results and outperform traditional ML models (Uchendu et al, 2021). Hence, the focus of this paper is transfer learning (of emotion) in a state of the art supervised detection method - which is fine-tuning an LLM. We want to use transfer learning to look directly at the impact of emotion in a direct head to head i.e. emoLLMsynth vs LLMsynth, so LLMsynth is our baseline.
>
>
> **Questions For The Authors:**
> >“What is the prompt used for data generation? Are data generated semantically correct?“
>
> An example prompt for text generation is given on lines 537-538 “In less than 400 words, tell me about moral philosophy.” An example prompt for the text classification task is given on lines 548-550 “Was the following written by a human or a computer, choose human or computer only?”
> The data generated by ChatGPT is semantically correct and was checked manually, the subject areas were chosen as the authors are knowledgeable in those areas and could check the correctness of the synthetic text. We will include this description in the paper.
> For Grover-generated synthetic text we use human written news articles as a prompt including the news article, headline, date, author, web domain etc. as described by Zellers et al (2019).
>
> >“LLMs models you [u]sed are very small and sentences without meaning are often generated. Can your results be influenced by this issue?”
>
> Some models are smaller, but we also use BERTlarge, BLOOM and ChatGPT for detection, and for generation we use Grover-base, Grover-mega, and ChatGPT.
> It is harder to detect larger LLMs, detection accuracy goes down for larger LLMs per results in Table 2 (Grover-base) and Table 3 (Grover-mega). For this reason, we use larger generator models and larger detector models and we look at trends across all these model sizes.
>
> Sentences without meaning could be a proxy which the LLM discriminator uses for classification, however, we account for this by comparing our emotionally-aware LLMs (emoLLMsynth) against standard LLMs fine-tuned on synthetic text detection only (LLMsynth). Thus, any advantage or disadvantage of sentences without meaning (or any other proxy) is also available to the non emotionally-aware model against which we compare our emotionally-aware model.
>
> >“Why do you not evaluate other simplest strategies for text classification as baseline?”
>
> As mentioned above, we focus on transfer learning using state of the art supervised detection i.e. fine-tuning a LLM, and our baseline is the fine-tuned LLM without emotion transfer learning i.e. LLMsynth.

---

### Official Review · Reviewer_8S6u · 2023-08-10

**Soundness:** 3

**Excitement:**

3: Ambivalent: It has merits (e.g., it reports state-of-the-art results, the idea is nice), but there are key weaknesses (e.g., it describes incremental work), and it can significantly benefit from another round of revision. However, I won't object to accepting it if my co-reviewers champion it.

**Missing References:**

(Minor) Ekman's 6 basic emotions are referenced many times throughout the paper, but the source is only referenced at the very end (limitations section). It would make sense to have that reference earlier (p. 3).

**Paper Topic And Main Contributions:**

The authors hypothesize that synthesized text, unlike human-generated text, lacks the implicit expression of emotion. Thus, they aim to exploit that quality to improve synthetic text classifiers.
The basic concept comprises three experimental steps: first, training a classifier on human and synthetic data (LLMsynth); second, training a classifier on emotion data (emoLLM); and third, an "emotionally-aware" classifier where the classification layer is replaced to be binary and trained on human vs synthetic data.
The training process and data acquisition are well-described. The composed data sets are part of the contribution. The trained models are compared and evaluated in different conditions, resulting in the overall conclusion that the emotionally-aware classifier indeed outperforms compared systems.


**Reasons To Accept:**

The paper is generally well-written and legible. Although the presented study has some weaknesses, it was accurately conducted and reported. The idea of exploiting emotional qualities to identify human-generated content is novel and exciting and addresses a current issue.

The drawn conclusion does not entirely convince me, but I believe this study provides some very interesting insights and is a valuable contribution to the overall discussion.
The released data sets are an additional contribution of this submission and might be relevant for future research.

**Reasons To Reject:**

The authors showcase the affective deficit in Sec. 3. However, the example rather shows a content-related incongruency (that would not happen to a human, but occasionally for machine-generated text) than a deficit of emotion words.
This example, in fact, underpins my main reservation: There are model improvements, but I am not entirely convinced that they can be traced to the emotional awareness of the model or if there might be any proxy. The authors address this issue and show the different characteristics of human and synthetic text in the appendix. There are some considerable differences in the data distribution, which could already be mentioned in Sec. 4.2. It can also be assumed that there are content differences.
Although the possibility of a proxy is discussed in the paper, it still needs to be solved and definitely remains for future research.
The results of the ChatGPT study are a nice addition but contribute little to the overall message of the paper.

**Reproducibility:**

4: Could mostly reproduce the results, but there may be some variation because of sample variance or minor variations in their interpretation of the protocol or method.

**Reviewer Confidence:**

4: Quite sure. I tried to check the important points carefully. It's unlikely, though conceivable, that I missed something that should affect my ratings.

**Typos Grammar Style And Presentation Improvements:**

I would prefer to refer to specific sections with "Sec. 5" instead of "§5". That might be a personal choice, but I feel like that's uncommon.

---

> ### Author Rebuttal · Authors · 2023-08-29
>
> Thank you for the feedback and suggestions.
>
> **Reasons to Reject:**
> We will clarify that the affective deficit is in the LLM, and this can result in affective incoherence in synthetic text generated by the LLM. We do not suggest that synthetic text contains no emotion, but that incongruencies can occur in text where emotion is being expressed. We will make this point clearer in the paper.
> We acknowledge that some other proxy could be at play, however in Table 5 the GNE vs GNEr results suggest emotion is playing a role because replacing the emotion detection task (GNE) with random labels (GNEr) causes a drop in accuracy.
>
> Thank you for the suggestion, we will include some examples where emoLLMsynth detected synthetic text that BERTsynth didn’t and there is affective coherence.
> * E.g. 1 "Marshal Yanda looked "**happy** and refreshed" before Wednesday's minicamp practice but made some notes. After a **disappointing** season, the 37-year-old Yanda might be "just getting **better and better and better**" physically, veteran linebacker Whitney Mercilus said.”
> * E.g. 2 "Bethenny Frankel, who hosted her daughter, Brynn, on ‘Bethenny Ever After,’ opened up about how she found balance when dealing with **two recent heartbreaks**. Bethenny Frankel, 39, has had a **rocky relationship** with her child, Bryn, 11, but all the drama behind the scenes has her engaged in **peace and happiness**.”
>
> Thank you for the suggestion, we will move Appendix E on sentence length etc. to §4.2
>
> The ChatGPT experiments were included to check performance of our emotionally-aware models in non-news domains, and test against a different synthetic text generator and detector (ChatGPT). In addition, with this paper we release the ChatGPT100 dataset which may be useful to other researchers.
>
> **Missing Refs:**
> Thank you for pointing this out, we will include the Ekman references where Ekman is first cited on p3.

---

### Meta-Review · Area_Chair_61pE · 2023-09-17

**Recommendation:** 3

**Metareview:**

The paper under discussion attempts to detect LLM synthetic texts. We hypothesize that LLM synthetic texts, compared to human-generated content, may lack emotion present. The authors hypothesize that this difference can be leveraged to create a more effective synthetic text classifier.

Pros:
1. All three reviewers acknowledge the novelty of the paper's idea – using the emotional difference between human and synthetic content to distinguish between them.
2. Clear writing and presentation. The proposed method is clearly described.
3. The datasets composed by the authors are a valuable contribution to the field and are likely to aid future research.

Cons:
1. Incomplete Analysis: Reviewer 1 and 3 highlights a main drawback in that there is insufficient analysis backing the central claim of the paper, particularly about the role of emotion in synthetic text detection.
2. Validation Concerns: Reviewer 2 brings up concerns about the lack of statistical validation. While the authors respond that the results in Table 2 are an average of five runs, it could be more robust if they reported variance or a significance test.

---

### Decision · Program_Chairs · 2023-10-07

**Decision:**

Accept-Findings

**Comment:**

The paper under discussion attempts to detect LLM synthetic texts. We hypothesize that LLM synthetic texts, compared to human-generated content, may lack emotion present. The authors hypothesize that this difference can be leveraged to create a more effective synthetic text classifier.

Pros:
1. All three reviewers acknowledge the novelty of the paper's idea – using the emotional difference between human and synthetic content to distinguish between them.
2. Clear writing and presentation. The proposed method is clearly described.
3. The datasets composed by the authors are a valuable contribution to the field and are likely to aid future research.

Cons:
1. Incomplete Analysis: Reviewer 1 and 3 highlights a main drawback in that there is insufficient analysis backing the central claim of the paper, particularly about the role of emotion in synthetic text detection.
2. Validation Concerns: Reviewer 2 brings up concerns about the lack of statistical validation. While the authors respond that the results in Table 2 are an average of five runs, it could be more robust if they reported variance or a significance test.